# Abundance and Distribution of *Phlebotomus pedifer* (Diptera: Psychodidae) Across Various Habitat Types in Endemic Foci of Cutaneous Leishmaniasis in the Mid-Highlands of Wolaita Zone, Southern Ethiopia

**DOI:** 10.3390/tropicalmed9120302

**Published:** 2024-12-10

**Authors:** Bereket Alemayehu, Temesgen Tomas, Negese Koroto, Teshome Matusala, Aberham Megaze, Herwig Leirs

**Affiliations:** 1Department of Biology, College of Natural and Computational Sciences, Wolaita Sodo University, Wolaita Sodo P.O. Box 138, Ethiopia; temesgen.tomas@bhu.edu.et (T.T.); negesekoroto6@gmail.com (N.K.); abreham.megaze@wsu.edu.et (A.M.); 2Evolutionary Ecology Group, Department of Biology, University of Antwerp, Campus Drie Eiken, Universiteitsplein 1, Wilrijk, 2610 Antwerp, Belgium; herwig.leirs@uantwerpen.be; 3Department of Geography, College of Social Sciences, Wolaita Sodo University, Wolaita Sodo P.O. Box 138, Ethiopia; teshome.matusala@wsu.edu.et

**Keywords:** phlebotomine sandflies, *Phlebotomus pedifer*, *Leishmania aethiopica*, leishmaniasis, habitat preferences, Wolaita Zone, Ethiopia

## Abstract

*Phlebotomus pedifer* is a vector of *Leishmania aethiopica*, the causative agent of cutaneous leishmaniasis. This study assessed the abundance and distribution of *P. pedifer* in different habitats and human houses situated at varying distances from hyrax (reservoir host) dwellings, in Wolaita Zone, southern Ethiopia. Sandflies were collected from January 2020 to December 2021 using CDC light traps, sticky paper traps, and locally made emergence traps. Sampling was performed in human houses, peri-domestic areas, farmlands, and hyrax dwellings. Houses 200 m and 400 m from hyrax dwellings were selected to study whether distance affects indoor sandfly abundance. A total of 2485 sandflies were captured, with *P. pedifer* accounting for 86.1% of the catch and *Sergentomyia* spp. comprising the remaining 13.9%. The abundance of *P. pedifer* was highest in human houses (72.3%) and lowest in farmlands (4.0%). Temperature showed a positive correlation with sandfly abundance (r = 0.434, *p* = 0.000), while rainfall (r = −0.424, *p* = 0.001) and humidity (r = −0.381, *p* = 0.001) were negatively correlated with abundance. Houses near hyrax dwellings had significantly higher *P. pedifer* abundance compared to those further away. Soil-emergence trapping yielded only a few *P. pedifer* specimens, primarily from hyrax dwellings. The findings highlight the increased presence of *P. pedifer* indoors, particularly in houses close to hyrax habitats, emphasizing the need for targeted indoor vector control strategies to mitigate the risk of cutaneous leishmaniasis transmission.

## 1. Introduction

Phlebotomine sandflies are small insects in the order Diptera, suborder Nematocera, family Psychodidae, and subfamily Phlebotominae [1]. There are over 800 species and subspecies of sandflies worldwide [2], of which over 90 species and subspecies are proven or suspected vectors for human leishmaniasis [3]. The vector species belong to either the genus *Phlebotomus* in the Old World or *Lutzomyia* in the New World [4]. In Ethiopia, over 22 species of *Phlebotomus* have been documented. Among these, *P. longipes* and *P. pedifer*, which belong to the subgenus *Larroussius*, are vectors of *Leishmania aethiopica*, the primary causative agent of cutaneous leishmaniasis (CL) in the country [5]. Other sandfly species, such as *P. sergenti* and *P. saevus* (vectors of *L. tropica*) and *P. duboscqi* (a vector for *L. major*), are also associated with CL transmission in the lowland areas of Ethiopia [5,6].

The ecology of vector sandflies in Ethiopia depends on the country’s diverse topography and climatic conditions, which vary across its agroecological zones. These zones are defined by altitude as follows: highland (“Dega”) at elevations above 2300 m above sea level (m a.s.l.), midland (“Woynadega”) ranging from 1500 to 2300 m a.s.l., and lowland (“Kola”) below 1500 m a.s.l. [7,8]. The vectors of visceral leishmaniasis are distributed in the lowland areas, while the vectors of CL are predominantly distributed in the highland and mid-highland areas in the country [9]. In general, the distribution and abundance of leishmania vectors are affected by abiotic factors (humidity, altitude, latitude, surface water, temperature, wind, and rainfall) and biotic factors (vegetation, parasites, competitors, predators, human interventions, and host species) [10]. In Ethiopia, the highland and mid-highland ecosystems provide better conditions for CL vectors and their associated reservoir hosts (hyraxes). Habitats in these ecologies are characterized by high-standing rock cliffs with fissures, sloppy areas with gorges and caves with dense vegetation for the habitation of hyraxes, and resting and breeding places for sandflies [11]. The ecological preference of leishmania vectors and reservoir hosts is one of the essential conditions influencing the dynamics of CL transmission in Ethiopia. For instance, the preferential distribution of the two confirmed vector sandfly species, *P. longipes* and *P. pedifer*, in the highland areas has limited CL transmission in the country’s highlands [12]. *Phlebotomus longipes* mainly transmits CL in the central and northern Ethiopian highlands, while *P. pedifer* is the main vector in the highlands of southern and southwestern Ethiopia [11].

The ecology and population dynamics of *P. pedifer*, have been extensively studied in Ochollo, a CL-endemic locality in the Gamo Zone in southwestern Ethiopia [9,11]. However, beyond Ochollo, detailed studies on the distribution and abundance of vector sandflies in southern Ethiopia remain limited. Such studies are greatly needed as CL is spreading to new areas in this part of the country. Information on CL vectors’ abundance, distribution, and habitat preference is very important to understand where and how human–CL transmission occurs. Therefore, this study aimed to assess the abundance and distribution of *P. pedifer* in various habitats and examine the variations in sandfly abundance within human houses located at different distances from the hyrax habitats.

## 2. Materials and Methods

### 2.1. The Study Area

This study was conducted in Damot Waja and Bosa Kacha villages, Sodo Zuria district, Wolaita Zone, southern Ethiopia (Figure 1). The district is located at 6°51′53′′ N and 37°45′44′′ E, with an elevation ranging from 1500 to 2958 m above sea level. The district has a total of 30 villages that are situated in two agroecologies, the highland (“Dega”) and the mid-altitude (“Woinadega”) [13]. The estimated population size of the district in 2023 was 215,129 people, out of which 105,785 were males and 109,344 were females [14]. Most communities in the district depend on farming as their primary source of income. The farmer communities predominantly cultivate maize, teff, barley, wheat, enset, banana, garlic, onion, sweet potato, potato, and sugarcane [15]. The study villages were chosen from among 30 in the Sodo Zuria district based on CL information provided by the Zonal Health Department.

Regarding the topography, the two villages are characterized by various land features, mostly mountain hills and plain surfaces. Of the two villages, Bosa Kacha has comparatively flat and plain surfaces and is affected by anthropogenic activities such as mining, deforestation, the rapid expansion of Wolaita Sodo town, and the release of the town’s waste. On the other hand, Damot Waja village is hilly and extends to the tip of Damota Mountain, which rises to an elevation of 2958 m a.s.l.

Concerning the area’s weather conditions, there is a bimodal rainfall pattern. The small rainy season occurs between March and May, followed by the main rainy season between June and September and a dry period between October and February. The mean monthly temperature ranges from 17.7 to 21.7 °C with an average of 19.7 °C [16].

### 2.2. Study Design

Based on the ecological information provided in previous related studies [9,11,17,18], a reconnaissance field survey was carried out, and suitable sandfly micro-habitats were searched to purposively select the habitat types in the study villages. To study the abundance and distribution of *P. pedifer* in habitat types, we set permanent trapping places in two hyrax habitats, two farmlands, two peri-domestic sites, and two human houses in each village. To study the effect of the distance between hyrax habitats and human houses on the abundance of sandflies, we selected three hyrax habitats, three households situated at 200 m from the hyrax habitats, and another three households at 400 m in each village. We used a GPS apparatus to select households and measure the specified aerial distance ranges from hyrax habitats. For trapping sandflies emerging from the soil, we selected two trapping places in each of the habitat types (hyrax habitat, farmland, peri-domestic, and human house).

### 2.3. Sandfly Collection

Adult sandflies were collected during three dry months (January, February, and December) and three wet months (June, July, and August), in both 2020 and 2021, for a total of twelve months. CDC miniature light traps (LTs) (John W. Hock Company, Gainesville, FL, USA) and sticky traps (STs) made of white laminated A4-sized papers coated with sesame oil were used for sandfly trapping. Sandflies emerging from the soil were trapped by using locally made emergence traps (ETs) that were constructed based on the methods of Casanova et al. [19] and Manteca-Acosta et al. [20], with modification. The adult sandfly traps were set for three consecutive nights every trapping month in four habitats: human houses, peri-domestic, farmlands, and hyrax habitats (rock fissures, boulders, and caves). A total of 1224 LTs and 6120 STs were deployed at 17 trapping places in each trapping village. An equal number of traps (one LT and five STs) were used per trapping place. The LTs were set at 6 p.m. and left overnight until 6 a.m., hanging approximately 1 m above the ground. The STs were placed in various forms, including rope-tying, pinning/leaning against walls, and bending/folding to fit within crevices. Sandflies trapped in LTs and STs were collected from the traps in the morning and preserved in 70% ethanol. The collected sandflies were transported to the Biomedical Science Laboratory of the Department of Biology, Wolaita Sodo University, to identify and examine the abdominal conditions of females (fed, unfed, and gravid). The head and the last two abdominal segments (in both sexes) were dissected and mounted in CMCP-10 high-viscosity mountant (Polysciences, Hirschberg, Germany). The mounted slides were examined through a microscope to identify the genera and species of the sandflies using relevant morphological keys [21,22,23]. Identification was based on the examination of key features located in the mounted head and abdomen, such as the cibarial and pharyngeal armature and genitalia [22,23].

To collect soil-emerging sandflies, 16 emergence traps consisting of oil-smeared plastic bottles fixed at the top were set at 8 places (2 at each habitat type: hyrax dwelling, farmland, peri-domestic, and human house). Soil-emerging sandflies, as they attempted to fly out, were trapped by the adhesive oil and subsequently collected using forceps. Each emergence trap was kept fixed on the ground (soil) for a duration of 42 days, which was based on the estimated time for adult sandflies to emerge [24,25]. The emergence traps were checked every three days to monitor the emergence of sandflies. After the completion of the entire trapping period, the trap was relocated to another suitable location within the same habitat to maximize the chances of trapping emerging sandflies.

### 2.4. Data Analysis

The data were coded, entered into an Excel spreadsheet, and subsequently imported into SPSS version 28 software for analysis. Sandfly abundance was descriptively presented as the total number of collections per study locality and habitat type. Before conducting the data analysis, the sandfly numbers were log-transformed using the formula log (*n* + 1) to fit a normal distribution. To verify data normality, we performed the Shapiro–Wilk test. Non-parametric tests were employed when the data did not meet the criteria for normal distribution. We used the Kruskal–Wallis (KW) test to observe the association of sandfly abundance among habitat types. When KW was significant, we performed the Mann–Whitney U test to determine the extent of the differences in the mean number of sandflies among habitat groups. The association between meteorological variables and the abundance of sandflies was evaluated using the Spearman correlation coefficient. All statistical tests were considered significant when the *p*-values were less than 0.05.

## 3. Results

### 3.1. The Overall Sandfly Abundance

Sandflies from two genera, *Phlebotomus* and *Sergentomyia*, were identified in the study (representative images of morphologically identified *P. pedifer* are available in Appendix A). A total of 2485 sandflies, 2140 (86.1%) *Phlebotomus pedifer* and 34 (13.9%) *Sergentomyia* spp., were collected by both LTs and STs. The numbers of sandflies collected from the two villages, Bosa Kacha and Damot Waja, were 1285 (51.7%) and 1200 (48.3%), respectively. The majority, 2320 (93.4%), of sandflies were collected by LTs, while only 165 (6.6%) were collected by STs. Of the total 2140 *P. pedifer*, 1165 (54.4%) were from Bosa Kacha village, and 975 (45.6%) were from Damot Waja village (Table 1).

### 3.2. Distribution of P. pedifer Among Habitat Types

It is important to note that the data presented in Table 2 exclude 1047 *P. pedifer* that were collected from habitats in the distance study because they could not be compared numerically due to differences in trapping habitats. Across various habitat types, we collected a total of 1093 *P. pedifer*, comprising 573 (52.4%) males and 520 (47.6%) females. These sandflies were trapped in human houses, peri-domestic areas, farmlands, and hyrax dwellings. We observed significant differences in the abundance of *P. pedifer* among these habitats (KW, χ^2^ = 42.631, df = 3, *p* = 0.001). In Bosa Kacha village, we trapped 513 sandflies, which accounted for about 46.9% of the total, while in Damot Waja village, there were 580 sandflies, making up about 53.1% of the total. Most of the sandflies, 78.6% in Bosa Kacha and 66.7% in Damot Waja, were collected from human houses. In contrast, the lowest proportion of sandflies was found in hyrax habitats (5.3%) in Bosa Kacha village and farmlands (1.4%) in Damot Waja village. The Mann–Whitney U test showed significant differences in the mean number of *P. pedifer* sandflies collected per trapping night between human houses and each of the other habitats (*p* < 0.05). In contrast, the mean number differences were not significant in comparisons made among peri-domestic areas, farmlands, and hyrax dwellings (Table 2).

### 3.3. P. pedifer Abundance with the Proximity of Households to Hyrax Dwellings

Out of the total *P. pedifer* captured for this specific study, 105 (10.0%) were collected from hyrax dwellings, 768 (73.4%) were from houses located 200 m away from hyrax dwellings, and 174 (16.6%) were from houses 400 m away in both villages. Of these collections, 612 (58.5%) were males, and 435 (41.5%) were females. There was a significant difference in the mean number of *P. pedifer* collected from houses at 200 m and 400 m from hyrax dwellings (*p* < 0.05) (Table 3).

### 3.4. Abdominal Conditions of the Female P. pedifer

From the total 520 female *P. pedifer* collected from the different habitat types, 343 (66.0%) were unfed, 139 (26.7%) were blood-fed, and 38 (7.3%) were gravid. The majority, 116 (83.5%), of the fed *P. pedifer* were trapped inside houses, followed by 10 (7.2%) from peri-domestic areas, 7 (5%) from hyrax dwellings, and 6 (4.3%) from farmlands. There was significant difference within the abdominal conditions of *P. pedifer* among different habitats (KW, χ^2^ = 59.753, df = 6, *p* = 0.000). Notably, a significant majority (90.1%) of the blood-fed sandflies were captured in houses located at a distance of 200 m from hyrax dwellings. In contrast, 6.4% were from houses at a 400 m distance, and 3.5% were from hyrax dwellings. The abdominal conditions of the sandflies also varied statistically with the proximity of houses to hyrax habitats (KW, χ^2^ = 37.572, df = 4, *p* = 0.001) (Table 4).

### 3.5. Seasonal and Meteorological Effects on the Abundance of P. pedifer

The abundance of *P. pedifer* showed distinct seasonal changes, with more collected during the dry months (January, February, and December), accounting for 779 (71.3%), compared to the wet months (June, July, and August) with 314 (28.7%). July had the lowest sandfly abundance, with only 79 (7.2%), while January had the highest density, with 386 (35.3%). This fluctuation in *P. pedifer* abundance correlated with temperature, showing a direct relationship. However, it had an inverse relationship with rainfall and relative humidity (Figure 2). Spearman’s correlation confirmed a positive association between *P. pedifer* and temperature (r = 0.434, *p* = 0.000), and a negative association with rainfall (r = −0.424, *p* = 0.001) and relative humidity (r = −0.381, *p* = 0.001) (Table 5).

### 3.6. Soil-Emerged P. pedifer Trapping

Only ten soil-emerged *P. pedifer* were trapped from soils at peri-domestic and hyrax dwelling sites, whereas none were trapped in human houses and farmlands. Six emerged sandflies were trapped at hyrax dwellings, while four were trapped at peri-domestic areas. The average number of days it took for sandflies to emerge was 38 (ranging from 34 to 40) (Table 6).

## 4. Discussion

*Phlebotomus pedifer* and *Sergentomyia* were collected from human houses, peri-domestic areas, farmlands, and hyrax habitats in Damot Waja and Bosa Kacha villages, in the Sodo Zuria district of Wolaita Zone, southern Ethiopia. Within the genus *Phlebotomus*, a single species, *P. pedifer*, was identified. Sandflies outside the *Phlebotomus* genus were not classified into specific species as they are not recognized vectors in Ethiopia. As the study villages are among the area’s CL endemic foci, our report focuses on the distribution and abundance of *P. pedifer*, the vectors of CL in the highlands and mid-highlands of southern Ethiopia [9,11]. *Phlebotomus pedifer* was the most abundant sandfly species in the current study area. Its higher abundance could be attributed to its altitudinal preference, as this sandfly species is typically found at altitudes ranging from 1600 to 2650 m above sea level [12,26]. Until our study, *P. pedifer* was primarily reported in Ochollo, a heavily researched CL-endemic area in southern Ethiopia [9,27,28,29]. These sandfly species tend to inhabit a range of habitats that are characterized by cooler temperatures, higher humidity, shade or darkness, and protection from wind [30]. They are often associated with vegetated rocky areas where hyraxes reside, such as caves, rock cliffs, and gorges [11]. Studies have shown that CL transmission rates increase when human populations settle close to these habitats [31]. While we did not test *P. pedifer* for *Leishmania* infection in this study, a limitation worth noting, a forthcoming study (manuscript under preparation) demonstrates PCR-confirmed *Leishmania* infection in *P. pedifer*, further substantiating its role in CL transmission within the endemic areas of Wolaita Zone.

The abundance of *P. pedifer* exhibited variations across the trapping habitats in the study villages, Damot Waja and Bosa Kacha. The highest proportion of *P. pedifer* was collected from human houses, whereas farmlands yielded the lowest proportion. This higher collection of sandflies indoors is likely due to adult sandflies entering homes from nearby breeding sites in search of blood meals or resting spots. Our observation of soil-emerging sandflies in peri-domestic locations implies that sandflies might be attracted to houses from the surrounding breeding areas. While our efforts to trap soil-emerging sandflies indoors were unsuccessful, a separate study has reported sandfly breeding on the earthen floors of human dwellings [32]. However, it is important to note that our study had limitations, such as setting emergence traps in only a few houses. Further investigations are required to ascertain whether sandflies can indeed breed indoors, especially in rural communities where humans and domestic animals share the same living spaces. Our findings contrast with some previous studies, which generally identify outdoor locations as the preferred habitats for sandflies, including *P. pedifer* [11,33,34,35,36]. For instance, our earlier study in the Kindo Didaye district of Wolaita Zone presented a significantly higher proportion of *P. pedifer* collected from hyrax dwellings [37]. In contrast, our study found a notably lower proportion of *P. pedifer* in hyrax habitats. These differences in sandfly proportions may be attributed to the specific microhabitats where the sandflies were trapped. The reduced sandfly collection from hyrax habitats in our study could result from the limited availability of rocky habitats or significant anthropogenic activities that may have disrupted the natural habitats of sandflies and their primary blood meal hosts, such as hyraxes. In general, the abundance of sandflies in outdoor areas is influenced by factors like soil type, weather conditions, animal dung, animal burrows, land features, land use, and the presence of mammalian hosts that serve as blood meal sources in the vicinity [32,35,38,39].

The two study villages, Damot Waja and Bosa Kacha, exhibited differences in *P. pedifer* abundance in their outdoor trapping habitats. Except for the farmlands, the other outdoor trapping habitats in Bosa Kacha village had fewer *P. pedifer* collections compared to Damot Waja village. Bosa Kacha village features fewer rocky areas, lacks caves, and has lighter vegetation coverage. These environmental characteristics, combined with environmental changes brought about by deforestation and the destruction of hyrax dwellings due to urbanization, may have contributed to the reduced number of sandfly collections, particularly from hyrax habitats. In contrast, the higher *P. pedifer* sandfly collection in the outdoor habitats of Damot Waja village might be attributed to the greater suitability of the habitat. Damot Waja village is a mountainous area that hosts several hyraxes in caves and rock outcrops.

In our study, the abundance of *P. pedifer* significantly varied among human houses located at different distances from the vegetated rocky habitats where hyraxes dwell. The higher abundance of sandflies observed in houses near hyrax habitats can be attributed to sandflies being more commonly found in houses close to their potential breeding and resting places. *Phlebotomus pedifer* has a limited flight range [40,41], which could further reduce their numbers in houses farther from their breeding and resting sites. Interestingly, our expectation of trapping the highest number of sandflies in hyrax dwellings, which are potential breeding and resting places, was not met. We found the lowest abundance of *P. pedifer* in hyrax habitats within the two study villages, a result that justifies further investigation. When comparing sandfly collections from houses situated at a distance (400 m away from hyrax habitats) in the two study villages, we found fewer sandflies in houses in Damot Waja village compared to those in Bosa Kacha village. The difference could be influenced by the more vegetated and rugged topography of Damot Waja village, potentially limiting sandfly dispersal. It is important to note that the flight range of sandflies may be affected by factors such as topography, land cover, and weather conditions [41,42,43]. Old-World *Phlebotomus* sandflies are known to be poor flyers and typically fly close to the ground in short hops [40,43].

The higher percentage of blood-fed *P. pedifer* trapped indoors in our study may be attributed to the behavior of these sandflies. *P. pedifer* are known to prefer indoor feeding (endophagic behavior) but can also be found both indoors and outdoors [28]. Research has shown that the females of these sandflies feed on a variety of vertebrate hosts, with a preference for hyraxes and humans as their primary sources of blood meals [11,28].

The abundance of *P. pedifer* exhibited seasonal fluctuations in our study. A significantly higher percentage of sandflies was collected during the dry season. While the number of sandfly collections decreased during the rainy months, there was no month without any trapping record, suggesting that *P. pedifer* was present throughout the year. The occurrence and abundance of *P. pedifer* during the dry season align with the findings from a study in Ochollo village in southern Ethiopia [9]. However, it is worth noting that the abundance may vary depending on the particular ecology and trapping method. Similar to other studies [9,35,44], our study also revealed a negative correlation between the abundance of *P. pedifer* and relative humidity and rainfall while showing a positive correlation with temperature.

Our efforts to locate breeding sites and trap *P. pedifer* in various habitats yielded limited results. Understanding the natural breeding sites of sandflies is a challenging task, as it involves time-consuming and complex efforts to locate their developmental stages [43,45]. We were able to capture only a small number of emerged sandflies from hyrax dwellings and peri-domestic areas, and our trapping attempts in other habitats were unsuccessful. Our findings are consistent with those of Foster [8] in Ethiopia, who indicated that caves and peri-domestic habitats were favorable breeding sites for *Phlebotomus* sandflies using soil extraction methods. Similarly, in Kenya, Mutinga and Odhiambo [33] confirmed that caves were the most suitable breeding sites for *P. pedifer*.

## 5. Conclusions

The present study revealed variations in the abundance of *P. pedifer* among indoor and outdoor habitats in the study districts. *P. pedifer* was more prevalent in human houses compared to other habitats. Notably, houses near hyrax dwellings had the highest abundance of this sandfly species, which could potentially increase the risk of households contracting CL. Additionally, this study highlighted a distinct seasonal pattern in *P. pedifer* abundance, with higher numbers during the dry season and a decrease during the wet season. While this research contributes to our understanding of *P. pedifer* ecology in the study areas, the relatively low abundance in hyrax habitats raises questions about the impact of human activities on these natural breeding and resting sites. The current higher abundance of *P. pedifer* in houses emphasizes the importance of implementing indoor vector control strategies in the study area.

## Figures and Tables

**Figure 1 tropicalmed-09-00302-f001:**
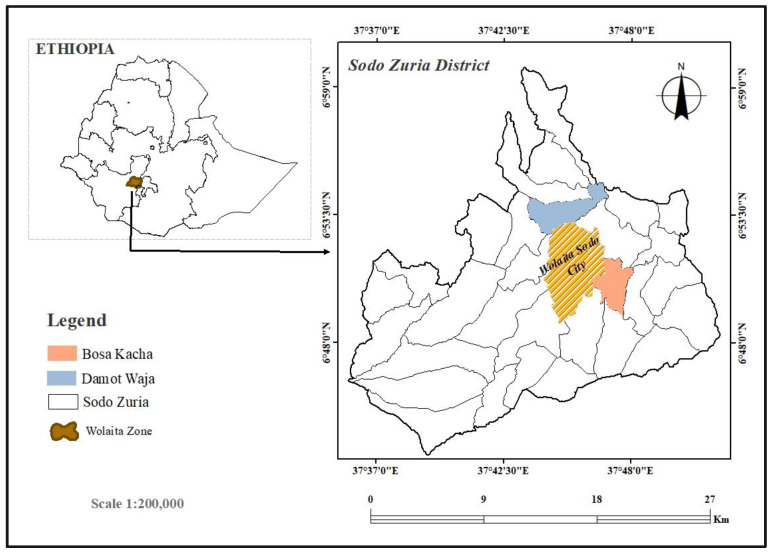
A location map of the study area (created with ESRI ArcGIS Desktop 10.8).

**Figure 2 tropicalmed-09-00302-f002:**
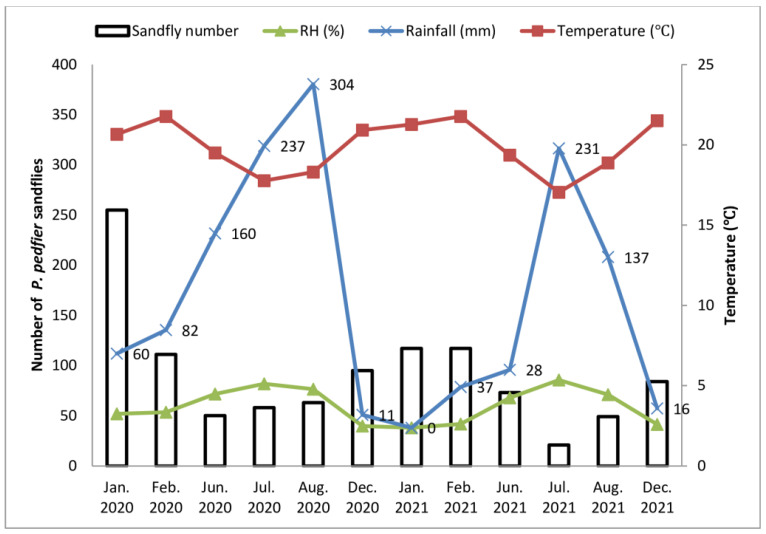
Abundance of *P. pedifer* with mean monthly temperature, relative humidity (RH), and rainfall, Wolaita Zone, southern Ethiopia.

**Table 1 tropicalmed-09-00302-t001:** Abundance of sandflies collected from Damot Waja and Bosa Kacha villages, Wolaita Zone, southern Ethiopia.

		Species of Sandflies		
		*P. pedifer*	*Sergentomyia* spp.	Total
Villages	Bosa Kacha	1165 (54.4%)	120 (34.8%)	1285 (51.7%)
	Damot Waja	975 (45.6%)	225 (65.2%)	1200 (48.3%)
Trap type	LT	2011 (94%)	309 (89.6%)	2320 (93.4%)
	ST	129 (6%)	36 (10.4%)	165 (6.6%)
Sex	Male	1185 (55.4%)	178 (51.6%)	1363 (54.8%)
	Female	955 (44.6%)	167 (48.4%)	1122 (45.2%)
Overall		2140 (86.1%)	345 (13.9%)	2485 (100%)

**Table 2 tropicalmed-09-00302-t002:** Abundance and sex distribution of *P. pedifer* in habitats in Damot Waja and Bosa Kacha villages, Wolaita Zone, southern Ethiopia.

	Damot Waja		Bosa Kacha		Total		χ^2^	df	*p*
	N (%)	^§^ Mean ± SE	N (%)	Mean ± SE	N (%)	Mean ± SE			
Habitats									
HHs	387 (66.7)	10.75 ± 4.29	403 (78.6)	11.2 ± 1.25	790 (72.3)	10.97 ± 2.5 ^a^	42.631	3	0.001
PD	81 (14.0)	2.25 ± 0.75	47 (9.2)	1.3 ± 0.56	128 (11.7)	1.78 ± 0.55 ^b^			
FL	8 (1.4)	0.22 ± 0.08	36 (5.0)	1 ± 0.31	44 (4.0)	0.6 ± 0.18 ^b^			
HDs	104 (17.9)	2.89 ± 0.95	27 (5.3)	0.75 ± 0.36	131 (12.0)	1.83 ± 0.56 ^b^			
Sex									
Male	347 (59.8)	2.4 ± 0.88	226 (44.1)	1.57 ± 0.36	573 (52.4)	2 ± 0.58	2.57	1	0.109
Female	233 (40.2)	1.6 ± 0.38	287 (55.9)	2 ± 0.41	520 (47.6)	1.8 ± 0.34			
Overall	580 (53.1)	4.03 ± 1.2	513 (46.9)	3.6 ± 0.73	1093 (100)	3.8 ± 0.88			

HHs: human houses, PD: peri-domestic, FL: farmlands, HDs: hyrax dwellings (habitats with actual hyrax presence), χ^2^: Kruskal–Wallis test, df: degree of freedom. Statistically significant differences (Mann–Whitney U test) are marked with different letters vertically. ^§^ mean number is calculated for sandfly collection per trapping night.

**Table 3 tropicalmed-09-00302-t003:** Abundance of *P. pedifer* sandflies between hyrax dwellings and houses at 200 m and 400 m distances from hyrax dwellings, Wolaita Zone, southern Ethiopia.

Trapping Sites	Bosa Kacha		D/Waja		Total		χ^2^	df	*p*
	N (%)	^§^ Mean ± SE	N (%)	^§^ Mean ± SE	N (%)	^§^ Mean ± SE			
HDs	38 (5.8)	1.06 ± 0.42	67 (17.0)	1.86 ± 0.72	105 (10.0)	1.46 ± 0.37 ^a^	23.552	2	0.001
HHs at 200 m	444 (68.1)	12.3 ± 1.48	324 (82.0)	8.97 ± 4.44	768 (73.4)	9.08 ± 1.21 ^b^			
HHs at 400 m	170 (26.1)	4.72 ± 0.65	4 (1.0)	0.1 ± 0.04	174 (16.6)	2.42 ± 0.34 ^a^			
Overall	652 (62.3)	6.04 ± 0.96	395 (37.7)	3.64 ± 1.6	1047 (100)	4.32 ± 0.71			

HH: human house, HDs: hyrax dwellings (habitats with actual hyrax presence), χ^2^: Kruskal–Wallis test, df: degree of freedom. Statistically significant differences (Mann–Whitney U test) are marked with different letters vertically. ^§^ mean number is calculated for sandfly collection per trapping night.

**Table 4 tropicalmed-09-00302-t004:** Abdominal conditions of *P. pedifer* collected from various habitat types, Wolaita Zone, southern Ethiopia.

Categories	Abdominal Conditions			Total N (%)	χ^2^	df	*p*-Value
	Unfed N (%)	Fed N (%)	Gravid N (%)				
Habitat types							
HHs	229 (66.8)	116 (83.5)	14 (36.8)	359 (69.0)	59.753	6	0.001
PD	56 (16.3)	10 (7.2)	2 (5.3)	68 (13.1)			
FL	14 (4.1)	6 (4.3)	6 (15.8)	26 (5.0)			
HDs	44 (12.8)	7 (5.0)	16 (42.1)	67 (12.9)			
Overall	343 (66)	139 (26.7)	38 (7.3)	520 (100)			
Distance from HD sites							
At HDs	30 (13.0)	6 (3.5)	3 (9.4)	39 (9.0)	37.572	4	0.001
HHs at 200 m	169 (73.1)	155 (90.1)	17 (53.1)	341 (70.4)			
HHs at 400 m	32 (13.9)	11 (6.4)	12 (37.5)	55 (12.6)			
Overall	231 (53.1)	172 (39.5)	32 (7.4)	435 (100)			

HHs: human houses, PD: peri-domestic areas, FL: farmlands, and HDs: hyrax dwellings (habitats with actual hyrax presence).

**Table 5 tropicalmed-09-00302-t005:** Correlation between the number of *P. pedfier* and meteorological factors such as temperature, relative humidity, and rainfall, Wolaita Zone, southern Ethiopia.

Meteorological Elements	Spearman’s Correlation	
	r-Value	*p*-Value
Temperature	r = 0.434	0.000
Relative humidity	r = −0.381	0.000
Rainfall	r = −0.424	0.000

**Table 6 tropicalmed-09-00302-t006:** Number of soil-emerged *P. pedifer* and trapping place characteristics in Wolaita Zone, southern Ethiopia.

Trapping Habitats	No of Emerged *P. pedifer* Trapped	Characteristics of Trapping Places
Human houses	0	Cracks on the walls, cracks on the floors chicken shed, cattle feces, animal barns
Peri-domestic areas	4 (40%)	Vegetation covers (coffee trees, “enset”, avocado trees, mango trees, and sugarcane), piles of household waste (animal dung and ash,) and plant litter
Farmlands	0	Several cracks in the soil
Hyrax dwellings	6 (60%)	Piles of hyrax feces in rock crevices and caves

## Data Availability

All relevant data are within the manuscript and its Appendix A files.

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
