# Peer review of "Abundance and Distribution of Phlebotomus pedifer (Diptera: Psychodidae) Across Various Habitat Types in Endemic Foci of Cutaneous Leishmaniasis in the Mid-Highlands of Wolaita Zone, Southern Ethiopia"

_tropicalmed, 2024, doi:10.3390/tropicalmed9120302_

Round 1

Reviewer 1 Report

Comments and Suggestions for Authors

Authors shall revise the manuscript more specifically

1. the grammar shall be revised

2. the method shall also be improved some how

3. The discussion section shall present the main result and its meaning. Then after, it can be compared with finding of similar reports.

Please find some comments attached to improve the manuscript.

Reviewer 2 Report

Comments and Suggestions for Authors

the ms presents data on distribution of sandflies in a region of Ethiopia. The data are of interest and are well presented. Please add images of the vectores as figures and discuss further on morphological ID since no molecular data are included.

Discussions section should be Discussion

Reviewer 3 Report

Comments and Suggestions for Authors

The authors present a very interesting and relevant study on the ecology of an endemic sandfly vector of a highly neglected parasite, namely Leishmania aethiopica, a zoonosis mainly of rock hyraxes in eastern Africa.

The manuscript is overall well written and compiled, the data, their analyses and the conclusions sound, except for a few points listed below.

Specific comments 

1.        Lines 36-37: Only generic and species names are to be formatted in italics, not higher ranks!!

2.        Line 41: Insert the information that both species belong to subgenus Larroussius.

3.        Lines 175-178: This information should appear at the start of section 3.2!!! In addition, the term "distant study" should be specified: do you mean the 200m vs 400m comparison? Then correct it to "distance study".

4.        Table 6, right column: Insert blank lines between the four categories/habitats to make reading easier.

5.        Line 249;  “These sandfly species…”  : Which ones??

6.        First paragraph of discussion: Please add a few words on the reason why neither in the present study nor in your previous papers treating the same focus, sandflies were PCR-tested for L.aethiopica rates and confirmation. This would have been particularly interesting in view of your sampling design, as you would have been able to evaluate the actual transmission risk with regard to habitat/distance et cetera. In a perfect scenario, you may already have these data and could mention them as “in prep.” (I do not suggest to incorporate them in the present MS as the subjects are too divergent). 

Reviewer 4 Report

Comments and Suggestions for Authors

Dear authors, I found on the website of Journal of Medical Entomology a published article by you (Bereket A., et al. 2024) entitled: The abundance and distribution of sand (with emphasis on Phlebotomus pedifer)  (Diptera: Psychodidae) along the altitudinal gradient in Kindo Didaye district, Wolaita zone, South Ethiopia. J. Med. Entomol. 61, 4: 940-947.

Both studies were carried out by similar or identical types of traps on the same habitats. Also, presented results in abstracts are very similar. I could only read the abstract chapter of this published paper, so I don’t have full insight into the whole context of this published paper. Furthermore, the paper published in J. Med. Entomology is not listed among references in references chapter of this submitted manuscript.

I suggest that the authors explain the differences between the proposed manuscript to journal of Tropical Medicine and Infectious Disease and the already published paper in Journal of Medical Entomology. For this reason I propose that this manuscript be rejected now, because without additional explanations, much remains in doubt. I think that double publication of similar or the same data should be avoided. If provided a valid explanation, you can resubmit this manuscript again.         

Round 2

Reviewer 4 Report

Comments and Suggestions for Authors

Dear Authors,

All improvements and corrections are adequate. 

Kind regards
